# Riemannian-Geometric Fingerprints of Generative Models

## Abstract

*Recent breakthroughs and rapid integration of generative models (GMs) have sparked interest in the problem of model attribution and their fingerprints. For instance, service providers need reliable methods of authenticating their models to protect their IP, while users and law enforcement seek to verify the source of generated content for accountability and trust. In addition, a growing threat of model collapse is arising, as more model-generated data are being fed back into sources (e.g., YouTube) that are often harvested for training ("regurgitative training"), heightening the need to differentiate synthetic from human data. Yet, a gap still exists in understanding generative models' fingerprints, we believe, stemming from the lack of a formal framework that can define, represent, and analyze the fingerprints in a principled way. To address this gap, we take a geometric approach and propose a new definition of artifact and fingerprint of GMs using Riemannian geometry, which allows us to leverage the rich theory of differential geometry. Our new definition generalizes previous work [60] to non-Euclidean manifolds by learning Riemannian metrics from data and replacing the Euclidean distances and nearest-neighbor search with geodesic distances and kNN-based Riemannian center of mass. We apply our theory to a new gradient-based algorithm for computing the fingerprints in practice. Results show that it is more effective in distinguishing a large array of GMs, spanning across 4 different datasets in 2 different resolutions (64×64, 256×256), 27 model architectures, and 2 modalities (Vision, Vision-Language). Using our proposed definition significantly improves the performance on model attribution, as well as a generalization to unseen datasets, model types, and modalities, suggesting its practical efficacy.*

## 1. Introduction

In recent years, we have seen a rapid development and integration of generative models (GMs) into our society. Vision generative models like Stable Diffusion [54] and Sora [7] are revolutionizing image and video synthesis, and Large-Language Models (LLMs) [1, 8] are changing how people work in business and science, including software development, entertainment, and scientific discovery. Despite this advancement, there is still a big gap in understanding what makes these models behave as they do and how one model differs from another. In particular, a rapidly arising question regarding GMs is model attribution and fingerprints, *i.e.*, what makes their synthetic data different from natural data (*e.g.*, GAN-generated images vs. real images), as well as from synthetic data generated by different models (*e.g.*, texts generated by GPT-4 [1] vs. by Gemini [62]). While this question has been partially studied in the context of deep-fake detection [65, 69], its full extension, *i.e.*, distinguishing *amongst* different methods of data generation (model attribution) remains mostly under-explored. In this paper, we address the problem of **fingerprinting and attributing generative models** in a theoretically-grounded way using Riemannian geometry [39] and manifold learning [3, 4, 23].

The problem of fingerprinting and attributing GMs bears increasingly critical significance in both practice and theory. First, in practice, service providers are looking for a reliable method of authenticating their proprietary models (*e.g.*, Google's Gemini [62], OpenAI's GPT [1]) to protect their IP, while users and law enforcement seek to verify the source of generated content for the trustworthiness of synthetic data and AI regulation [42, 51]. In addition, a growing threat of model collapse [13, 59] is arising as more model-generated data are being fed back into sources (*e.g.*, YouTube) that are often harvested for training ("regurgitative training"), heightening the need to detect and differentiate synthetic from human data. Secondly, on the theoretical front, model attribution and fingerprints provide a formal, analytical way of studying the differences between various GMs and revealing their unique characteristics and limitations, thereby promoting the development of new models that overcome current limitations (*e.g.*, artifact-reduced GMs [10, 21, 30, 58]).

In this paper, we study the problem of fingerprinting GMs based on their samples and provide a formal framework that can define, represent, and analyze the fingerprints to study and compare GMs in a principled way. Recent works on deepfake detection [49, 56] and model biases [58, 65] have suggested that GMs leave distinct traces of computations on

Figure 1. **Overview of our fingerprint estimation.** We learn the latent data manifold $\mathcal{M}$ from the dataset of real images as a Riemannian manifold equipped with the pullback metric $G$. To pullback the metric of $\mathcal{X}$ to the latent manifold $\mathcal{M}$, we train a VAE (with mean and variance estimation functions $(\theta_\mu, \theta_\sigma)$) on the real dataset, and define the length of a curve on $\mathcal{M}$ to be the length of a decoded curve on $\mathcal{X}$, on which we know how to measure a length (*i.e.* a Euclidean norm of a vector). From the length of a curve, we define the geodesic distance of two points on $\mathcal{M}$ as the shortest distance of a curve connecting them.

their samples that differentiate model-generated data from human data (*e.g.*, GM-generated images vs. images by cameras). Such traces have been observed in both the pixel space (*e.g.*, checkerboard patterns by deconvolution layers [50], semantic inconsistencies like asymmetric eye colors [47]) and the frequency space (*e.g.*, spectral discrepancies [15, 16]). Despite these observations that hint at the existence of artifacts and fingerprints of GMs, an explicit definition of fingerprints themselves remains unclear. A recent work [60] proposes the first definition of fingerprints, but its applicability to real-world data is limited due to its assumption that the embedding space of data is Euclidean, which is often violated in real data like images and videos which follow non-Euclidean geometry [6]. This lack of a proper definition hinders a systematic study of GM fingerprints (that goes beyond showing their mere existence), and development of fingerprinting methods for real-world model attribution.

To this end, the aim of this work is to (i) give a proper definition of GM fingerprints that generalizes to **non-Euclidean** spaces using **Riemannian geometry**, (ii) apply our theory to a new gradient-based algorithm for computing the fingerprints, by learning Riemannian metrics from data and replacing the Euclidean distances and nearest-neighbor search with geodesic distances and $k$NN-based Riemannian center of mass, and finally (iii) study their efficacy in differentiating a large variety of GMs and generalizing across datasets, model types and modalities. We find that our proposed definition provides a useful feature space for fingerprinting GMs, including state-of-the-art (SoTA) models, and outperforms existing methods on both attribution and generalization.

By providing a formal definition of GM fingerprints, we address another important gap in literature, *i.e.*, the lack of studies on fingerprints across different modalities. While SoTA GMs are being developed on multimodal data (*e.g.*, a combination of images, texts and audios), studies on GM fingerprints have been limited to a single-modality (*e.g.*, images only [60, 65, 69] or texts only [68]). To bridge this gap and encourage research on cross-modal fingerprints, we introduce an extended benchmark dataset (Tab. 2) that includes SoTA multimodal GMs (*i.e.*, text-to-image models). Our contributions can be summarized as following:

- We formalize a definition of fingerprints of GMs that generalizes to non-Euclidean data using Riemannian geometry.
- We propose a practical algorithm to compute the fingerprints from finite samples by learning a latent Riemannian metric as pullback metric from the data space and estimating the fingerprints via a gradient-based algorithm.
- We conduct extensive experiments to show the attributability and generalizability of our fingerprints, outperforming existing attribution methods. In particular, we consider a large array of GMs from all four main families (GAN, VAE, Flow, Score-based), spanning across 4 different datasets (CIFAR-10 [37], CelebA-64 [40], CelebA-HQ [29] and FFHQ [31]) of 2 different resolutions ($64\times64$, $256\times256$), 27 model architectures, and 2 modalities (Vision, Vision-Language). This, to our knowledge, is the most comprehensive model-attribution study to date.
- Our results show that our generalized definition makes significant improvements on attribution accuracy and generalizability to unseen datasets, model types and modalities, suggesting its efficacy in real-world scenarios.

## 2. Riemannian-Geometric Fingerprints of GMs

As discussed in Sec. 1, explicit formal definitions of artifacts and fingerprints of GMs remain unclear or constrained to a specific data geometry despite many existing works that

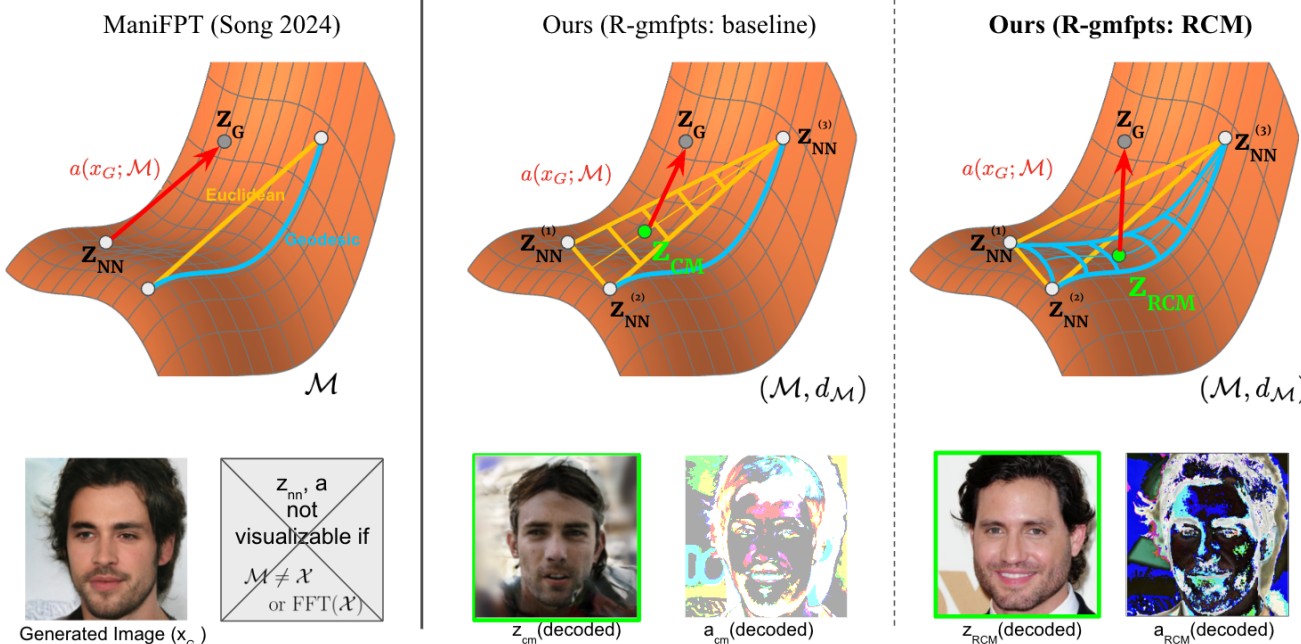

Figure 2. **Estimating the projection of $x_G$ onto the manifold $\mathcal{M}$ as Riemannian center of mass of $k$-nearest neighbors.** *(left)* Definition and fingerprint estimation proposed in [60]. Here, the projection of $z_G$ is estimated the nearest-neighbor (1-NN) in the observed real dataset, based on the standard Euclidean distance. *(middle)* Baseline method using $k$-nearest neighbors ($k>1$; $k=3$ in this figure): we estimate the artifact of $z_G$ by finding the $k$-nearest neighbors of $z_G$ in the real dataset using the Euclidean distance, and computing their center of mass (also in L2). *(right)* Our proposed method (**R-gmftps: RCM**) that estimates the artifact $a(x_G, \mathcal{M})$ using a Riemannian center of mass of $k$-nearest neighbors, based on the geodesic distances learned from data (Sec. 2.2). Note $z_{cm}$ does *not* lie on the manifold $\mathcal{M}$ (which is manifested on the synthetic artifacts in $z_{cm}$ (decoded) in the center), while the projection $z_{RCM}$ estimated as Riemannian center of mass (right column) *does* lie on $\mathcal{M}$, thus corresponding to an actual real image. (Background manifold image modified with permission [23])

observed their existence across various classes of GMs. In particular, the first definition of GM fingerprints proposed in [60] assumes the data manifold to be Euclidean, but such assumption is often violated in the high-dimensional data we encounter in the real world (*e.g.*, images, videos, and texts).

Motivated by this limitation, in this section, we propose an improved formal definitions of artifacts and fingerprints of generative models that generalize to non-euclidean data by taking into account their geometry. We then describe a practical algorithm for computing them from observed samples, by (i) learning a Riemannian metric from data as a pullback metric [3, 4], and (ii) estimating the projection of generated samples to this learned manifold as a Riemannian center of mass (RCM) [2] of $k$-nearest neighbors in geodesics.

## 2.1. Definitions of GM artifacts and fingerprints

The artifacts and fingerprints of generative models intuitively correspond to the models' defects in matching the generative process of real data, which are manifested consistently in the samples they generate. As per the manifold learning hypothesis [9, 18], which assumes real-world data, including images and texts, lie on a lower-dimensional manifold, we can formalize such deficiencies of GMs as how their gener-

ated samples deviate from the true data manifold. Formally, let $G$ be a generative model trained on the dataset $X_R$ of real samples that lie on a lower-dimensional data manifold $\mathcal{M}$, $P_G$ its induced probability distribution, $S_G$ its support, and $x_G$ its sample:

**Definition 2.1** (Artifact). An artifact left by generative model $G$ on its sample $x_G$ (denoted as $\boldsymbol{a(x_G; \mathcal{M})}$) is defined as the difference between $x_G$ and its projection $x^*$ onto the manifold $\mathcal{M}$ of the real data used to train $G$:

$$x^* := \text{Projection}(x_G, \mathcal{M}) \tag{1}$$

$$\boldsymbol{a(x_G; \mathcal{M})} := x_G - x^* \tag{2}$$

**Definition 2.2** (Fingerprint). The fingerprint of a generative model $G$ with respect to the data manifold $\mathcal{M}$ is defined as the set of all its artifacts over its support $S_G$:

$$\textbf{Fingerprint}(G; \mathcal{M}) = \{\boldsymbol{a(x_G; \mathcal{M})} | x \in S_G\} \tag{3}$$

Figure 1 (left) illustrates our proposed definitions.

## 2.2. Estimation of Riemannian GM artifacts and fingerprints

Our estimation of GM artifacts and fingerprints consist of two main steps: (i) We first learn the latent data manifold $\mathcal{M}$

from the dataset of real images as a Riemannian manifold equipped with proper metric tensor $g$. (ii) We then estimate the artifact on $x_G$ according to this metric, by computing the projection of $x_G$ onto $\mathcal{M}$ as a Riemannian Center of Mass (RCM) of $x_G$'s $k$-nearest neighbors in $\mathcal{M}$, and the artifact as the difference between $x_G$ and its projection on $\mathcal{M}$. Finally, the fingerprint of $G$ is computed as the set of artifacts for each $x_G$ in its sample set $X_G$. Algorithm 1 describes the workflow of our fingerprint estimation (notations in Tab. 1).

---

**Algorithm 1** Compute Riemannian GM fingerprints

**Input:** Set of real images ($X_R$) and images generated by model $G$ ($X_G$)

1: **function** COMPUTE-FINGERPRINT($X_G \mid X_R$)
2:      **Fingerprint$(G; R)$** $\leftarrow \{\}$
3:      **for** $x_G \in X_G$ **do**     ▷ Runs in parallel
4:          $a(x_G) \leftarrow$ COMPUTE-ARTIFACT($x_G \mid X_R$)
5:          **Fingerprint$(G; R)$**.add($a(x_G)$)
6:      **return** **Fingerprint$(G; R)$**

7: **function** COMPUTE-ARTIFACT($x \mid X_R$)
8:      $(\mathcal{M}, g) \leftarrow$ LEARN-RIEMANNIAN-MANIFOLD($X_R$)
        ▷ Learned data manifold with metric tensor $g$
9:      $x^\star \leftarrow$ PROJECT($x, \mathcal{M}$)    ▷ Project $x$ onto $\mathcal{M}$
10:      $a(x, \mathcal{M}) \leftarrow x - x^\star$      ▷ Artifact as a difference vector
11:      **return** $a(x, \mathcal{M})$

---

**Step 1. Learning the Riemannian manifold from data.**
Since we do not have access to the data manifold $\mathcal{M}$ on which the real images lie (*i.e.*, the natural image manifold), we need to estimate it using the observed samples at hand. To this end, we use real images in the training datasets of the generative models, and map them to a suitable embedding space to construct a collection of features to be used as an estimated image manifold. Unlike the previous definition in [60], which used either the pixel space, its FFT space, or the embedding spaces of pretrained networks (ResNet and Barlow-Twin pretrained on ImageNet), we *learn* the latent data manifold from the observed real dataset.

The key idea in our approach to learning the latent data manifold from the observed real dataset (*e.g.*, real images) with metric is to pull back the geometry in the observation space (*e.g.*, L2 distances in the image pixel space) to the latent manifold, using generative models. This metric learning based on a pullback is proposed in [3, 4]. In our work, we choose as the generative model a VAE with the mean and variance estimators and train it on the real dataset with the standard VAE loss function. Its learned encoder functions as the embedding map from the data space to the latent space, and its decoder as a vehicle to define the geometry of the

latent space (*e.g.*, length of a curve and geodesic distances on the manifold) by pulling back the geometry of the data:**(i).** Train VAE with mean and variance parameters $(\theta_\mu, \theta_\sigma)$ with the standard VAE loss, using the real dataset $X_R$: Note that *both* the mean and variance parameters are required for proper learning of Riemannian metrics [4, 23], and we train such VAE by (i) first training its inference network with the variance parameter fixed, and (ii) training the variance parameter as in [4] (Sec 4.1). This step provides the embedding map $f_{enc} : \mathcal{X} \rightarrow \mathcal{M}$, and $f_{dec} : \mathcal{M} \rightarrow \mathcal{X}$.

**(ii).** Define the metric on $\mathcal{M}$ by pulling back the metric on $\mathcal{X}$ (*i.e.*, L2 in the standard generative modeling setup [35]), to $\mathcal{M}$: To do so, we first define the "length" of a curve in M to be the "length" of its decoded curve on X, *i.e.*,

$$\text{length}_{\mathcal{M}}(c) := \text{length}_{\mathcal{X}}(f_{\text{dec}}(c)) \quad (4)$$

and define the pullback metric as the metric tensor $g$ on $\mathcal{M}$:

$$g := J_\mu^T J_\mu + J_\sigma^T J_\sigma \ [4] \quad (5)$$

**Step 2. Estimating the artifact of $G$ in its sample.** The artifact of $G$ in its sample $x_G$ is computed in two steps:
1. Estimate the projection $x^\star$ as the RCM of $x_G$'s $k$-nearest neighbors on the data manifold:
   - Compute $k\text{NN}_{d_{\mathcal{Z}}}(x_G, \mathcal{M})$, the set of $k$-nearest neighbors ($k$NNs) on the data manifold, according to the distance metric $d_{\mathcal{Z}}$ of $\mathcal{Z}$
   - Estimate $x^\star$ as RCM($k\text{NN}_{d_{\mathcal{Z}}}(x_G, \mathcal{M})$): compute the Riemannian center of mass of the $k$NNs using the metric on the latent Riemannian manifold, learned in Step.1 (see Alg. 2).
2. Compute the artifact as a difference vector between $x_G$ and $x^\star$: $a(x_G; \mathcal{M}) = x_G - x^*$

Algorithm 3 describes how to compute the projection of $x_G$ onto $\mathcal{M}$ using its $k$-nearest neighbors of $x_G$ in $\mathcal{M}$ and the Riemannian center of mass of the $k$NNs.

**Computing Riemannian center of mass (RCM) on the latent manifold.** Please see details on how we define and compute the RCM on the latent manifold in B.2.

**Step 3. Computing the fingerprint of a generative model.** Given a set of model-generated samples $X_G = \{x_i\}_{i=1}^N$ where $x_i \sim P_G$, we estimate its fingerprint w.r.t. real data $X_R$ by computing an artifact of each sample in $X_G$, *i.e.* **Fingerprint**$(G; X_R) = \{a(x; X_R) \mid x \in X_G\}$.

### 2.3. Attribution network

We train a ResNet50-based classifier to predict source models given our artifact representations. See C for details.

## 3. Experiments

We show that using our proposed definition significantly improves the performance on model attribution and zero-shot generalization to unseen datasets, models and modalities. Please see E for details on our experiments.

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

# A. Background on Riemannian geometry and Riemannian manifold learning

To establish a common language and context for introducing our Riemannian-geometric definition of fingerprints, we start with a brief recap of Riemannian geometry [19, 39, 57].

**Riemannian manifold and its metric.** A Riemmanian manifold is a well-studied metric space that locally resembles a Euclidean space, allowing geometric notions such as lengths, distance and curvature to be (locally) defined on a curved (*i.e.* non-Euclidean) space. A formal definition is as follows:

**Definition A.1.** A Riemmanian manifold is a smooth manifold $\mathcal{M}$, whose tangent space $T_p\mathcal{M}$ at each point $p \in \mathcal{M}$ is equipped with an inner product (Riemannian metric) $g_p : T_p\mathcal{M} \times T_p\mathcal{M} \to \mathbb{R}$ in a smooth way.

**Riemannian metric.** A Riemannian metric $g$ on $\mathcal{M}$ assigns to each $p$ an inner product $g_p : T_p\mathcal{M} \times T_p\mathcal{M} \to \mathbb{R}$, which induces a norm (*i.e.*, length of a vector) $||\cdot||_p : T_p\mathcal{M} \to \mathbb{R}$ defined by $||v||_p = \sqrt{g_p(v,v)}$.

Inituitively, a Riemannian metric functions as a measuring stick on every tangent space of $\mathcal{M}$, dictating how to measure the lengths of vectors and curves (and other derived geometric quantities) on $\mathcal{M}$.

**Geometric calculations on Riemannian manifolds.**

- Measuring lengths on a Riemannian manifold: The inner product structure on $(\mathcal{M}, g)$ is sufficient for defining length of a smooth curve $c : [0, 1] \to \mathcal{M}$ as:

$$\text{Length}(c) := \int_0^1 \sqrt{g_{c(t)}(\dot{c}(t), \dot{c}(t))}\mathrm{d}t \qquad (6)$$

where $\dot{c} = \partial_t c$ denotes the curve velocity.

- Measuring distances on a Riemannian manifold: Given $p, q \in \mathcal{M}$, the distance from $p$ to $q$ is defined as:

$$\mathrm{d}(p, q) := \inf\{\text{Length}(c)|c : [0, 1] \to \mathcal{M}, \qquad (7)$$
$$c(0) = p, c(1) = q\}$$

Such a minimum-distance curve from $p$ to $q$ is called a "geodesic" and its distance the "geodesic distance".

**Learning Riemannian manifold from data.** In the context of this paper we focus on Riemannian manifolds whose structure and Riemannian metrics are *learned* from observed data (*i.e.*, images). In particular, we employ the method of *pulling back* the metric on the observed data space to the latent manifold [3, 4, 23] by training a generative model (*e.g.*, VAE) with its encoder and decoder functions.

More specifically, given the data space $\mathcal{X}$ and a VAE $(f_{\text{enc}}, f_{\text{dec}})$ trained with a latent space $\mathcal{Z}$, where its decoder function $f_{\text{dec}}$ models *both* the mean and variance, *i.e.*,

$$f_{\text{dec}}(z) = \mu(z) + \sigma(z) \odot \epsilon, \qquad (8)$$
$$\mu : \mathcal{Z} \to \mathcal{X}, \sigma : \mathcal{Z} \to \mathbb{R}_+, \epsilon \sim \mathcal{N}(0, \mathbb{I}_D)$$

| | |
|---|---|
| $(\mathcal{X}, d_{\mathcal{X}})$ | Observed data space as $(\mathbb{R}^D, d_{\text{L2}})$ |
| $(\mathcal{Z}, d_{\mathcal{Z}})$ | Latent space of VAE as $(\mathbb{R}^d, d_{\text{L2}})$ |
| $(\mathcal{M}, d_{\mathcal{M}})$ | Latent Riemannian manifold immersed in $\mathcal{X}$ with $d_{\mathcal{M}} := g_{\text{pullback}}$ s.t. $g_p : T_p\mathcal{M} \times T_p\mathcal{M} \to \mathbb{R}$ |
| VAE | Generative model that learns a latent manifold of real dataset; $(f_{\text{enc}}, f_{\text{dec}})$ |
| $X_R, X_G$ | real dataset, and synthetic dataset generated by a generative model $G$ |
| $x_{\text{NN}}^{(k)}$ | $k$th nearest neighbors of $x$ on the manifold $(\mathcal{M}, d_{\mathcal{M}})$ |
| $k\text{NN}(x; X_R)$ | the set of $K$ nearest neighbors of $x$ on $(\mathcal{M}, d_{\mathcal{M}})$ |
| $x_{\text{RCM}}$ | Riemannian center of mass of $k\text{NN}(x; X_R)$ on $(\mathcal{M}, d_{\mathcal{M}})$ |
| $\boldsymbol{a}(\boldsymbol{x_G}; \boldsymbol{\mathcal{M}})$ | Artifact of $x_G$ with respect to $(\mathcal{M}, d_{\mathcal{M}})$ |

Table 1. Our notation: data and latent spaces as metric spaces and points involved in estimating GM fingerprints.

we can construct a Riemmanian metric $g$ on $\mathcal{Z}$ from the metric on $\mathcal{X}$ (which is, conveniently, Euclidean), as [4]:

$$g := \mathbf{J}_\mu^T\mathbf{J}_\mu + \mathbf{J}_\sigma^T\mathbf{J}_\sigma \qquad (9)$$

where $\mathbf{J}_\mu$ and $\mathbf{J}_\sigma$ are the Jacobians of $\mu(.)$ and $\sigma(.)$. These Jacobians can be estimated from the standard backpropagation through the trained decoder [4].

In our approach (Sec. 2), we use this pullback approach to learn a latent (Riemannian) geometric structure of real image datasets, and estimate "artifacts" of a GM on model-generated data using this learned, real data manifold as a reference of a true data distribution. Additionally, the learned metric aids in computing a more geometrically appropriate notion of "projection" to the manifold, using geodesic distances and Riemannian center of mass. We now introduce our new definition and implementation of GM fingerprints.

# B. Our Method: Riemannian-Geometric Fingerprints of GMs

## B.1. Algorithms for computering fingerprints

---

**Algorithm 4** Compute RCM: Gradient Descent for finding the Riemannian $L^p$ center of mass of $\{x_i\}_{i=1}^N$

---

**Input:** $\{x_i\}_{i=1}^N \subset \mathcal{M}$, $\{w_i\}_{i=1}^N$ and choose $x^0 \in \mathcal{M}$
1: `if` $\nabla f_p(x^k) = 0$ `then` `stop`, `else` `set`

$$x^{k+1} \leftarrow \exp_{x^k}(-t_k\nabla f_p(x^k)) \qquad (10)$$

where $t_k > 0$ is an "appropriate" step-size and $\nabla f_p(\cdot)$ is defined in (12).
2: `repeat` step 1

---

---

**Algorithm 2** Learn Riemannian manifold from real dataset

---

**Input:** Set of real images, $X_R$

1: **function** LEARN-RIEMANNIAN-MANIFOLD($X_R$)
2:     # Train VAE with mean and variance parameters $(\theta_\mu, \theta_\sigma)$ using $X_R$ with standard VAE losses.
3:     **VAE** = $(f_{\text{enc}}, f_{\text{dec}}) \leftarrow$ TrainVAE($X_R$)    $\triangleright f_{\text{enc}}$ : $\mathcal{X} \to \mathcal{Z}, f_{\text{dec}} : \mathcal{Z} \to \mathcal{X}$
4:     # Pullback metric from the data space $\mathcal{X}$ to $\mathcal{M}$
5:     $\boldsymbol{g} \leftarrow J_\mu^T J_\mu + J_\sigma^T J_\sigma$    $\triangleright$ Riemannian metric on $\mathcal{M}$, pulled back through $f_{dec}$
6:     $\mathcal{M} \leftarrow f_{\text{enc}}(X_R)$
7:     **return** $(\mathcal{M}, \boldsymbol{g}), \textbf{VAE} = (f_{enc}, f_{dec})$

---

**Algorithm 3** Compute projection of $x_G$ onto $\mathcal{M}$ as RCM

---

**Input:** A model-generated sample $x_G$, and Riemannian manifold $\mathcal{M}$

1: **function** PROJECT($x_G, \mathcal{M}$)
2:     # Compute the projection of $x_G$ onto $\mathcal{M}$ as RCM of $k$NNs in $\mathcal{M}$
3:     $\boldsymbol{Q}_K \leftarrow k\text{NN}_{\boldsymbol{d}_{\boldsymbol{z}}}(x, X_R)$    $\triangleright k$-nearest neighbors of $x$ on $\mathcal{M}$ using the metric $d_{\mathcal{Z}}$
4:     **RCM**($\boldsymbol{Q}_K$) $\leftarrow$ COMPUTE-RCM($Q_K, G$) $\triangleright$ Riemannian center of mass of $k$NNs on $\mathcal{M}$
5:     **return** **RCM**($\boldsymbol{Q}_K$)

---

### B.2. Computing Riemannian center of mass (RCM) on the latent manifold.

We first define the RCM as following [2]:

**Definition B.1.** Given the dataset $\{x_i\}_{i=1}^N \subset \mathcal{M}$ and $L^p$-distance $d^p$ on $\mathcal{M}$, its Riemannian $L^p$ center of mass (a.k.a. Fréchet mean) with respect to weights $0 \leq w_i \leq 1$ ($\sum_{i=1}^N w_i = 1$) is defined as the minimizer(s) of

$$f_p(x) = \begin{cases} \frac{1}{p} \sum_{i=1}^N w_i d^p(x, x_i) & 1 \leq p < \infty \\ \max_i d(x, x_i) & p = \infty \end{cases} \quad [2] \quad (11)$$

in $\mathcal{M}$. By convention, the weights are assumed to be equally distributed unless specified [2]. Our method adopts an iterative, gradient-based algorithm [2, 38, 43] for computing this Riemannian center of mass, using the metric $g$ learned in Step 1 (Eqn. 5). Alg. 4 describes how we compute a local RCM given a set of query points $Q \subset \mathcal{M}$ on our learned data manifold $\mathcal{M}$, using the pullback metric $g$ as the distance $d$ in Eq. 11. In particular, we follow until convergence the gradient of $f_p$ using the $k$-nearest neighbors of $x_G$ on $X_R$:

$$\nabla f_p(x) = - \sum_{i=1}^K w_i d^{p-2}(x, x_i) \exp_x^{-1} x_i \ [2] \quad (12)$$

This gradient is valid for any $x \in \mathcal{M}$ as long as it is not in the small open neighborhood of other data points [2]. For our purposes, we assume that this condition is achieved due to the sparsity of our observed sample, *i.e.*, the dimensionality of our data manifold (*i.e.*, image manifold) is much higher than our sample size (*i.e.*, number of observed images). In Algo. 4, we initialize $x^0$ with a random data point in $X_R$, and set $t_k$ for each $k$ as in [38].

## C. Method: Attribution network

Our attribution network takes as input our artifact representation of an image (computed in Step 2) and predicts the identity of its source generative model: First, we represent the input image as an artifact feature by computing its deviation from the learned data manifold, as discussed in Sec. 2.2. Note that we do not learn the Riemannian manifold (and its metric) at the inference time, since this step is done only once per given real data during training or data-preprocessing steps. Next, the artifact feature is fed into our ResNet-based classifier to perform multi-class classification over the generative models. To train this classifier, we use the pretrained ResNet50 [25] as the backbone and finetune it with the cross-entropy loss accruing from classifying images in the training split of our dataset in Tab. 2.

## D. Experiments: Dataset Creation

ManiFPT [60] provides an extensive benchmark dataset for evaluating model attribution across a large array of GMs, spanning 4 different training datasets and all 4 main GM familys (GAN, VAE, Flow, Diffusion). However, what it is currently lacking is the inclusion of multimodal models such as vision-language models. To bridge this gap, and to evaluate model attribution methods on a wider variety of models, we created an extended benchmark dataset that includes SoTA vision-language GMs. In particular, we include 4 SoTA models (last row of Tab. 2) that can generate images given input text prompts: Flux.1-dev, Stable-Diffusion-3.5, Dall-E-3, and Openjourney. For all these models, we used pre-trained models that are available either on Huggingface or on public Github repositories.

### D.1. Details on dataset creation

**GM-CelebA dataset.** To construct a dataset of faces that resemble images in CelebA [40], we use the text prompt of "a face of celebrity" to each of the vision-language models. For example, for Flux.1-dev model, we use the Huggingface's 'diffuser' library to download the model weights, and used each pretrained model with default sampling configurations to generate 10k images with this prompt.

**GM-CIFAR10 dataset.** To generate images like the data in CIFAR10, we created a text prompt for each class in CIFAR10 (*i.e.*, airplane, automobile, bird, cat, deer, dog,

| Family | GM-CIFAR10 | GM-CelebA | GM-CHQ | GM-FFHQ |
|---|---|---|---|---|
| Real | CIFAR-10 [37] | CelebA [40] | CelebA-HQ (256) [29] | FFHQ (256) [31] |
| GAN | BigGAN-Deep [5] | plain GAN [20] | BigGAN-Deep [5] | BigGAN-Deep [5] |
| | StyleGAN2 [32] | DCGAN [53] | StyleGAN2 [32] | StyleGAN2 [32] |
| | | LSGAN [44] | StyleGAN3 [30] | StyleGAN3 [30] |
| | WGAN-gp [22] | WGAN-gp/lp [22] | VQ-GAN [17] | VQ-GAN [17] |
| | | DRAGAN-gp/lp [36] | StyleSwin [70] | |
| | DDGAN [67] | | DDGAN [67] | |
| VAE | | $\beta$-VAE [26] | | |
| | | DFC-VAE [28] | StyleALAE [52] | |
| | NVAE [63] | NVAE [63] | NVAE [63] | NVAE [63] |
| | VAE-BM [66] | VAE-BM [66] | VAE-BM [66] | |
| | Eff-VDVAE [24] | Eff-VDVAE [24] | Eff-VDVAE [24] | Eff-VDVAE [24] |
| Flow | GLOW [34] | GLOW [34] | | |
| | MaCow [41] | | MaCow [41] | |
| | | | Residual Flow [11] | |
| Score | DDPM [27] | DDPM [27] | DDPM [27] | |
| | | | NCSN++ [61] | NCSN++ [61] |
| | RVE [33] | RVE [33] | RVE [33] | |
| | LSGM [64] | | LSGM [64] | |
| | | | LDM [55] | LDM [55] |
| Vision-Language | Flux.1-dev | Stable-Diffusion-3.5 | Dall-E-3 | Openjourney |

Table 2. **Our experimental dataset of generation models.** We introduce an extended benchmark dataset for model attribution that includes SoTA multimodal GMs (*i.e.*, text-to-image models) in addition to the large array of SoTA GMs trained on 4 different datasets (CIFAR10, CelebA, CelebA-HQ(256), FFHQ(256)) in 2 different resolutions (64×64, 256×256) from [60]. We evaluate the model fingerprints on their attributability and cross-data/model/modality generalization. **Real**: training datasets of the generative models. **Score**: score-based (a.k.a. diffusion) models.

frog, horse, ship, truck ), as "an image of {cifar10-class}". We then provided this prompt to each of the vision-language models we added in Tab. 2, and used each pretrained model with its default sampling configurations to generate a total of 10k images per prompt per CIFAR10 class.

## E. Experiments

We evaluate our proposed fingerprint definitions and attribution method on model attribution and generalization to unseen datasets, generative models and modalities. Sec. E.1 explains our experimental setup. Sec. E.2 and Sec. E.3 evaluate the attributability of GM fingerprints on a large array of generative models via (multi-class) model attribution and feature space analysis. Sec. E.4 studies the generalizability of our fingerprints across dataset, model type, and modalities.

### E.1. Experimental setup

**Datasets.** To evaluate the performance of different fingerprints on model attribution for a multi-class classification, we use the GM datasets from [60] – GM-CIFAR10, GM-CelebA, GM-CHQ and GM-FFHQ – constructed from real datasets and generative models trained on CIFAR-10 [37], CelebA-64 [40], CelebA-HQ(256) [29] and FFHQ(256) [31], respectively. This dataset includes 100k images from each generative model, collectively covering GAN, VAE, Flow, and diffusion families, and state-of-the-art generative models (*e.g.*, NAVE, NCSN++, LSGM) that have not been considered before. In addition, we extend this dataset to include

SoTA multimodal GMs (vision-language models) to evaluate fingerprints' generalization across a wider variety of modalities. Tab. 2 summarizes our datasets, organized in column by the training datasets, with the last row indicating the vision-language models. We emphasize that each dataset in the column exclusively consists of images from models trained on the same training dataset, which is crucial for evaluating the effects of model architectures and datasets on attribution and cross-dataset generalization independently, which we study in Sec. E.2 and Sec. E.4.

**Baselines** We consider the three main categories of existing model attribution methods: color-based, frequency-based and supervised-learning methods. We evaluate representative methods from each group and compare them to our proposed method. Additionally, we include the SoTA fingerprinting method based on a data manifold (ManiFPT [60]) in our comparison.

- Color-based methods: Histogram of saturated and underexposed pixels [47], Color co-occurrence matrix [48]
- Frequency-based methods: 1-dim power spectrum via azimuthal integration on DCT [15], high-frequency decay parameters fitted to normalized reduced spectra [12, 16]
- Supervised learning methods: InceptionNet-v3 [45], XceptionNet [45], Yu et al. [69], Wang et al. [65]
- Manifold-based methods (Euclidean): $\text{ManiFPT}_{\text{RGB}}$, $\text{ManiFPT}_{\text{FREQ}}$ [60] which use RGB and frequency (FFT) spaces, respectively, as their embedding spaces

For comparison, we consider two variants of our attribution method, **R-GMfpts** which computes the RCM on $\mathcal{M}$ using Euclidean distances, and **R-GMfpts$_{\text{RCM}}$** which uses the learned geodesic distances.

### E.2. Attribution of generative models

We test the attributability of fingerprints by training a classifier for model attribution based solely on our computed artifacts. For other baselines, the inputs to the classifier may be a full image or frequency spectrum, depending on their expected representations. A high test accuracy implies the classifier is able to predict source models from the fingerprints, thereby supporting the existence of their fingerprints.

**Metrics.** We evaluate the attributability of the baseline methods listed in Sec. E.1 and our proposed methods on our GM datasets (Tab. 2). The performance is measured in classification accuracy (%).

**Evaluation Protocol.** Each dataset in Tab. 2 consists of real images and synthetic images from GMs trained on those real images. Each image is labeled with the ID of its source model (*e.g.*, 0 for Real, 1 for $G_1$, ..., M for $G_M$). We split the data into train, val, test in ratio of 7:2:1, train the methods on the train split and measure the accuracies on the test split.

**Results.** Tab. 3 shows the result of model attribution. First, we observe that our attribution methods (**R-GMfpts**, **R-GMfpts$_{\text{RCM}}$**) outperform all compared methods on all

| Methods | GM-CIFAR10 | | GM-CelebA | | GM-CHQ | | GM-FFHQ | |
|---|---|---|---|---|---|---|---|---|
| | Acc.(%)↑ | FDR↑ | Acc.(%)↑ | FDR↑ | Acc.(%)↑ | FDR↑ | Acc.(%)↑ | FDR↑ |
| McCloskey et al. [47] | $40.22 \pm 1.10$ | 32.4 | $62.6 \pm 0.31$ | 70.2 | $57.4 \pm 0.81$ | 36.3 | $50.8 \pm 0.34$ | 26.3 |
| Nataraj et al. [48] | $46.29 \pm 1.43$ | 36.7 | $61.1 \pm 0.91$ | 74.0 | $56.3 \pm 0.32$ | 37.9 | $51.3 \pm 0.58$ | 35.3 |
| Durall et al. [15] | $57.29 \pm 0.93$ | 46.5 | $62.2 \pm 0.24$ | 75.5 | $59.1 \pm 0.80$ | 38.8 | $60.9 \pm 0.25$ | 37.9 |
| Dzanic et al. [16] | $56.12 \pm 1.21$ | 43.1 | $61.6 \pm 1.02$ | 88.1 | $56.9 \pm 1.21$ | 38.2 | $55.7 \pm 0.32$ | 30.3 |
| Corvi et al. [12] | $60.12 \pm 0.14$ | 47.2 | $59.2 \pm 0.59$ | 46.5 | $60.5 \pm 0.76$ | 48.2 | $59.3 \pm 0.53$ | 49.2 |
| Wang et al. [65] | $62.23 \pm 0.84$ | 53.6 | $62.2 \pm 1.20$ | 89.8 | $59.5 \pm 1.25$ | 30.3 | $64.2 \pm 0.31$ | 37.9 |
| Marra et al. (MIPR) [45] | $55.94 \pm 1.09$ | 41.2 | $63.1 \pm 1.10$ | 83.4 | $51.3 \pm 1.28$ | 20.5 | $53.2 \pm 0.21$ | 30.4 |
| Marra et al. (WIFS) [46] | $60.71 \pm 1.24$ | 47.2 | $61.1 \pm 1.72$ | 101.4 | $59.1 \pm 0.75$ | 34.9 | $51.8 \pm 0.23$ | 30.9 |
| Yu et al. [69] | $62.01 \pm 0.79$ | 50.1 | $60.6 \pm 1.10$ | 111.4 | $61.1 \pm 1.12$ | 73.3 | $60.5 \pm 0.10$ | 35.1 |
| ManiFPT$_{RGB}$ [60] | $69.48 \pm 1.08$ | 55.2 | $70.5 \pm 1.56$ | 115.3 | $63.7 \pm 0.63$ | 64.2 | $63.3 \pm 0.12$ | 50.1 |
| ManiFPT$_{FREQ}$ [60] | $70.19 \pm 0.96$ | 57.2 | $72.8 \pm 1.32$ | 120.9 | $54.8 \pm 0.32$ | 70.1 | $63.8 \pm 0.20$ | 43.8 |
| **R-GMfpts (ours)** | $\underline{72.01} \pm 0.92$ | $\underline{58.9}$ | $\underline{73.6} \pm 0.70$ | **168.0** | $\underline{64.3} \pm 0.72$ | **77.2** | $\underline{65.2} \pm 0.30$ | $\underline{58.8}$ |
| **R-GMfpts$_{RCM}$ (ours)** | **78.17** $\pm 0.53$ | **60.1** | **74.7** $\pm 0.62$ | $\underline{125.9}$ | **65.8** $\pm 0.75$ | 74.5 | **67.1** $\pm 0.20$ | **60.8** |

Table 3. **Model attribution results.** We evaluate different artifact features on predicting the source generative model of a generated sample. Separability of the feature spaces are measured in Fréchet distance ratio (FDR). Higher FDR means better separability. Our methods based on the proposed definition of artifacts outperform all baseline methods on all datasets (results from [60] and our evaluations).

datasets with significant margins. In particular, our method achieves higher accuracies across all dataset than the Euclidean-based method (ManiFPT [60]), supporting that taking the Riemannian geometry structure of data when estimating the GM fingerprints improves their estimation.

Secondly, we note that our method that takes full advantage of the learned Riemannian metric on the data manifold (**R-GMfpts$_{RCM}$**) outperforms the one that does not and uses a standard Euclidean metric instead (**R-GMfpts**). This result suggests that computing the projection of $x_G$ as a Riemannian center of mass using the learned geodesics contributes the estimation to lie closer to the data manifold, minimizing potential errors in the artifact computation. We visualize this point in Figure 3 (center vs. right) where $z_{CM}$ (center) lies off the manifold whereas $z_{RCM}$ (right) on the manifold.

### E.3. Feature space analysis

**Separability (FD ratio).** We measure the separability of fingerprint representations using the ratio of inter-class and intra-class Fréchet Distance (FDR) [14] as done in [60]. The larger the ratio, the more attributable the fingerprints are to their source models. Tab. 3 shows the FD ratios computed for the fingerprints on the test datasets. We observe that the FDRs are significantly higher for learned representations (Row of Wang et al. and below) than color-based (McCloskey [47], Nataraj [48]) and frequency-based discrepancies (Durall [15], Dzanic [16], Corvi [12]). In particular, the feature spaces based on our new Riemannian definitions achieve improved FDRs, in alignment with the attribution results in classification accuracy.

| Methods | C10→CA | CA→C10 | CHQ→FFHQ | FFHQ→CHQ |
|---|---|---|---|---|
| McCloskey et al. [47] | $52.3 \pm 0.14$ | $43.2 \pm 0.08$ | $34.2 \pm 0.13$ | $31.2 \pm 0.10$ |
| Nataraj et al. [48] | $56.2 \pm 0.11$ | $46.1 \pm 0.19$ | $42.1 \pm 0.07$ | $40.4 \pm 0.19$ |
| Durall et al. [15] | $60.1 \pm 0.14$ | $53.5 \pm 0.12$ | $51.9 \pm 0.09$ | $42.6 \pm 0.12$ |
| Dzanic et al. [16] | $56.9 \pm 0.13$ | $54.7 \pm 0.11$ | $45.2 \pm 0.22$ | $42.5 \pm 0.17$ |
| Corvi et al. [12] | $59.2 \pm 0.21$ | $59.3 \pm 0.14$ | $48.1 \pm 0.35$ | $46.6 \pm 0.14$ |
| Wang et al. [65] | $62.5 \pm 0.15$ | $60.1 \pm 0.10$ | $61.4 \pm 0.13$ | $53.4 \pm 0.12$ |
| Marra et al. (MIPR) [45] | $57.0 \pm 0.14$ | $58.4 \pm 0.33$ | $50.2 \pm 0.17$ | $35.9 \pm 0.27$ |
| Marra et al. (WIFS) [46] | $61.0 \pm 0.15$ | $58.6 \pm 0.23$ | $54.3 \pm 0.12$ | $30.3 \pm 0.17$ |
| Yu et al. [69] | $60.5 \pm 0.13$ | $60.4 \pm 0.20$ | $55.2 \pm 0.17$ | $50.3 \pm 0.13$ |
| ManiFPT$_{RGB}$ [60] | $62.7 \pm 0.14$ | $60.2 \pm 0.15$ | $58.1 \pm 0.22$ | $53.2 \pm 0.11$ |
| ManiFPT$_{FREQ}$ [60] | $65.8 \pm 0.12$ | $62.1 \pm 0.11$ | $57.6 \pm 0.20$ | $53.5 \pm 0.18$ |
| **R-GMfpts (ours)** | **68.2** $\pm 0.11$ | $\underline{62.3} \pm 0.16$ | $\underline{63.5} \pm 0.14$ | **56.9** $\pm 0.28$ |
| **R-GMfpts$_{RCM}$ (ours)** | $\underline{67.8} \pm 0.21$ | **65.0** $\pm 0.12$ | **67.3** $\pm 0.24$ | $\underline{54.3} \pm 0.32$ |

Table 4. **Generalization of model attribution across datasets.** We evaluate how well baselines and our fingerprints generalize across training datasets. We consider two scenarios: (i) generalization across GM-CIFAR10 and GM-CelebA, and (ii) generalization across GM-CHQ and GM-FFHQ. For each case, we train attribution methods on one set of generative models (*e.g.*, GM-CIFAR10) and test on a different set of models (*e.g.*, GM-CelebA). Our artifact-based attribution method outperforms all baseline methods in both scenarios. **C10**: CIFAR-10. **CA**: CelebA. **CHQ**: CelebA-HQ.

### E.4. Cross-dataset generalization

We evaluate the generalizability of fingerprinting methods across datasets. Since new models are developed constantly by training or fine-turning on new datasets, this cross-dataset generalizability is crucial in practice. We consider two scenarios whose data semantics are meaningfully different: generalization (i) between GM-CIFAR10 and GM-CelebA, and (ii) between GM-CHQ and GM-FFHQ.

**Evaluation.** In each case, we train attribution methods on the training dataset (*e.g.*, GM-CelebA) and test their accuracies on the other unseen dataset (*e.g.*, GM-CIFAR10).

**Results.** Tab. 4 shows the result of cross-dataset generalizations for GM-CIFAR10 ↔ GM-CelebA and for GM-CHQ ↔ GM-FFHQ. First, our attribution methods based

on Riemannian fingerprints (**R-GMfpts$_{RCM}$**, **R-GMfpts**) outperform all compared methods in both cases of cross-generalization. Note that CIFAR-10 and CelebA contain images from different domains (CIFAR-10: objects and animals vs. CelebA: human faces).Therefore, the high accuracies in this particular scenario indicate that our new fingeprints generalize not only across datasets of similar semantics (CHQ $\leftrightarrow$ FFHQ), but also across of different semantics (CIFAR-10 $\leftrightarrow$ CelebA). Overall, these higher accuracies show that our methods are more generalizable in the midst of the changes of training datasets of various generative models, supporting the efficacy of our methods in practice where attribution is needed to address end users who can train new models using their own datasets.

## F. Conclusion

Our work addresses an increasingly critical problem of attributing and fingerprinting GMs, by proposing fingerprints that generalize to non-Euclidean data using Riemannian geometry. Our experiments on an extended SoTA dataset showed that our method is more effective in distinguishing a wider variety of GMs (4 datasets in 2 resolutions, 27 model architectures, 2 modalities). Using our definition significantly improved performance on model attribution and generalization. We believe this generalized definition of GM fingerprints can help address gaps in the theory and practical understanding of GMs, by providing a geometric framework to systematically analyze the characteristics of GMs.

