# OpenReview forum: "Riemannian-Geometric Fingerprints of Generative Models"
_thecvf.com/ICCV/2025/Workshop/BEW — BEW 2025 Abstract_

### Official Review · Reviewer_BD2u · 2025-07-01
**Good submission**

**Rating:** 5
**Confidence:** 5

**Review:**

The work proposes a new approach to fingerprinting of generative models that alleviates the Euclidean assumption on the embedding space in prior work. In particular, it leverages Riemannian geometry and defines an artifact as the difference between a sample and its projection onto the manifold that represents the real data. Further, fingerprints are then defined as the set of artifacts over a given support set. Leveraging this definition in context of a recent gradient-based algorithm for computing fingerprints and learning Riemannian metrics from the data as well as replacing Euclidean distances with geodesic distances, the authors demonstrate as part of a very extensive empirical evaluation that their proposed is more robust and provides better model attribution.

Some additional discussion/hypothesis on the euclidean version of R-GMfpts outperforming the one using the Riemannian metric in some cases would have been beneficial to further strengthen the experimental discussion.

The proposed approach is novel, addresses a highly relevant problem and provides extensive empirical results to support the proposed methodology. The work is overall presented in a very clear manner when disregarding the somewhat abrupt jump from main paper to the appendix due to the page-limit. Overall, I believe it will make a nice contribution to the workshop.

---

### Official Review · Reviewer_6bXs · 2025-07-03

**Rating:** 5
**Confidence:** 4

**Review:**

This paper proposes a novel, geometry-based framework for analyzing and attributing generative models by defining their fingerprints using Riemannian geometry. Unlike prior approaches that assume Euclidean data spaces, the authors introduce a method that learns a latent Riemannian manifold from real data using a variational autoencoder (VAE), and defines artifacts as deviations of generated samples from this manifold. Fingerprints are then formalised as the collection of such artefacts.

To compute these fingerprints, the authors develop a gradient-based algorithm that estimates projections onto the learned manifold using Riemannian centres of mass based on geodesic distances. The resulting fingerprint representations are used to train a ResNet-based classifier for model attribution.

Several experiments demonstrate that this Riemannian approach significantly outperforms existing attribution methods, both in classification accuracy and feature space separability. The method also generalises well to unseen models and datasets.

While the method is computationally intensive and relies on the quality of the learned VAE, it provides a principled and effective framework for GM fingerprinting and attribution.

Main is issue is that the paper has no conclusion.

---

### Decision · Program_Chairs · 2025-07-09

**Decision:**

Accept (Abstract)

**Comment:**

The reviewers agree that this paper will be an interesting contribution for the poster discussion during the workshop.